# BCL-2 Expression in AML Patients over 65 Years: Impact on Outcomes across Different Therapeutic Strategies

**DOI:** 10.3390/jcm10215096

**Published:** 2021-10-30

**Authors:** Mario Tiribelli, Angela Michelutti, Margherita Cavallin, Sara Di Giusto, Erica Simeone, Renato Fanin, Daniela Damiani

**Affiliations:** 1Division of Hematology and Stem Cell Transplantation, Azienda Sanitaria Universitaria Friuli Centrale—Ospedale S. M. Misericordia, 33100 Udine, Italy; mario.tiribelli@uniud.it (M.T.); angela.michelutti@asufd.sanita.fvg.it (A.M.); margherita.cavallin@asufd.sanita.fvg.it (M.C.); digiustosara@gmail.com (S.D.G.); erica.simeone@asufd.sanita.fvg.it (E.S.); renato.fanin@uniud.it (R.F.); 2Department of Medical Area, University of Udine, 33100 Udine, Italy

**Keywords:** BCL-2, acute myeloid leukemia, elderly, prognosis, survival

## Abstract

BCL-2 overexpression has been associated with resistance to chemotherapy and reduced survival in acute myeloid leukemia (AML), but few data are available in elderly patients, a subset accounting for majority of AML cases and with dismal prognosis. We retrospectively analyzed 113 AML patients aged ≥65 years treated with 3 + 7 chemotherapy (*n* = 51) or hypomethylating agents (HMAs) (*n* = 62), evaluating the role of BCL-2 expression on complete remission (CR) and overall survival (OS). BCL-2 was expressed in 81 patients (72%), more frequently in those with unfavorable cytogenetic-molecular risk. CR was achieved in 34.5% cases, without differences according to BCL-2 expression or induction therapy. In the whole population 1-year OS was 39%, similar in BCL-2+ and BCL-2- cases. In BCL-2 positive patients OS was superior with HMAs (56% vs. 25% with 3 + 7; *p* = 0.02), while no advantage for HMA was found in BCL-2 negative cases (36% vs. 27% for 3 + 7). Therapy with HMAs was the only factor associated with longer OS in BCL-2+ AML by multivariable analysis. Use of HMAs, possibly in combination with BCL-2 inhibitors, appears to be particularly appealing in BCL2+ AML, where it is associated with superior survival.

## 1. Introduction

The incidence of acute myeloid leukemia (AML) increases with age, and majority of AML patients are older than 65 years [1]. Despite improvements in chemotherapy and supportive care, the prognosis for elderly AML patients remains dismal, due to more aggressive disease biology and unfitness to intensive chemotherapy. BCL-2 protein plays a major role in regulation of cell death mechanisms, including apoptosis, necrosis and autophagy, is frequently overexpressed in AML, and is associated with chemoresistance and shortened overall survival [2,3,4]. Since BCL-2 expression has been linked to both disease pathogenesis and resistance to therapy, the concept of acting on its function is an appealing strategy in the design of novel anti-leukemic approaches. Venetoclax (AbbVie, Lake Bluff, Illinois, USA) (formerly ABT-199) is an oral inhibitor of BCL-2 that induced potent and selective apoptosis in AML cell lines and patient samples, as well as in xenograft murine models [5]. US Food and Drug Administration (FDA) have approved venetoclax, in combination with the hypomethylating agent (HMA) azacytidine or low doses cytarabine (LDAC), for the treatment of AML patients who are older or unfit for intensive chemotherapy.

DiNardo and colleagues had reported on the efficacy of a combination of venetoclax and HMAs (decitabine or azacytidine) in a cohort of 145 treatment-naïve AML patients at least 65 years old. [6]. They reported a high complete remission (CR) rate at 67% and a median overall survival (OS) of 17.5 months, with favorable responses also in high-risk subgroups, such as over 75 years, poor-risk cytogenetic, and secondary AML. However, it was not reported on a possible correlation between BCL-2 expression and response to the combination of venetoclax and HMAs, despite the same group had previously shown that increased BCL-2 expression is associated with higher sensitivity to venetoclax [5].

The aim of our study was to retrospectively analyze impact of BCL-2 expression on the outcome of a cohort of elderly AML patients, treated with intensive chemotherapy or HMAs, and the association with other clinical and laboratory characteristics possibly identifying subsets of patients taking higher advantage from BCL-2 inhibition.

## 2. Methods

### 2.1. Patients

A total of 113 patients with non-promyelocytic AML and age ≥65 year referred to the Division of Hematology and Stem Cell Transplantation of Udine University Hospital from January 2010 to December 2020 were retrospectively included in this study. Median age was 68 years (range: 65–85), 49 patients (43%) had de novo AML, and 64 (57%) had secondary AML. Cytogenetic-molecular risk was classified according to 2010 ELN recommendations [7].

### 2.2. BCL2 Expression

BCL-2 expression on leukemic cells was tested at diagnosis by flow cytometry using the anti-human BCL-2 Oncoprotein FITC (clone 124), Agilent Dako, Santa Clara, CA, USA); the monoclonal antibody was employed in accordance with the manufacturer’s guidelines. Briefly, 1 × 10^6^ cells were incubated for 10 min at room temperature with FACS Lysing Solution (Becton Dickinson) to permeabilize the cell membrane, washed, and incubated at room temperature with 10 μL of the antibody in a 0.02% saponin solution for 15 min. After two washes cells were stained with 10 μL of anti-human CD34PE (clone 8G12, Becton Dickinson) and 10 μL of anti-human CD45 PerCP-Cy5.5 PE (clone 2D1, Becton Dickinson) for 15 min at room temperature, washed again, and analyzed. Anti-human CD33 PE (clone p67.6) was used in CD34 negative samples. Blast cell population was first identified according to CD45 expression and forward scatter characteristics, then BCL-2 expression was evaluated in CD34 positive cells.

For BCL-2 intensity of expression evaluation, the ratio between the mean fluorescence channel of test and that of the BCL-2 isotypic control was calculated (an example of the analysis is shown in Appendix A). Results were expressed as the mean fluorescence index (MFI) by calculating the ratio between the mean fluorescence intensity of cells that were incubated with the anti-human BCL-2 oncoprotein and the mean fluorescence intensity of the respective isotypic control. BCL-2 was considered overexpressed for MFI ≥ 17 (i.e., above the median value of the population).

### 2.3. Therapy

Fifty-one patients (45%) received induction chemotherapy with standard 3 + 7 idarubicin and cytarabine chemotherapy, 62 patients (55%) were treated with HMAs (decitabine 10 days/cycle *n* = 37, azacytidine 7 days/cycle *n* = 25). None of the 64 patients with secondary AML had previously received HMA therapy for MDS. Patients attaining complete remission (CR) with standard chemotherapy received at least one consolidation course with 3 + 7 or intermediate dose cytarabine. No patient underwent allogeneic hematopoietic cell transplant. CR was defined as the complete hematological recovery and a bone marrow blast count <5% by morphologic and immunophenotypic evaluation.

### 2.4. Response Definition

Overall survival (OS) was defined as time from diagnosis to death, irrespective of its cause. Leukemia-free survival (LFS) was defined as the time from the date of CR to the date of relapse. Patients lost to follow-up were censored at the time last seen alive.

### 2.5. Statistical Analyses

Categorical variables were compared with the Fisher exact test or Yates corrected chi square test, as required. Factors affecting CR were assessed by univariate and multivariate logistic regression and expressed as HR (95% CI). Survival curves were constructed by the Kaplan–Meier method and differences among groups calculated by the log-rank test. The Cox proportional hazard regression model was used to examine the potential prognostic factors for survival: All variables with *p* values ≤ 0.10 in univariate analysis were included in the multivariable model and a forward procedure was applied to identify significant factors. Statistics was performed by NCSS 11 Statistical Software (2018) (NCSS, LLC. Kaysville, UT, USA, ncss.com/software/ncss, accessed on 24 February 2021). All *p*-values are 2-sided at a significance level of 0.05.

## 3. Results

### 3.1. BCL2 Expression and Correlation with Other Features

High BCL-2 expression (BCL-2+) was found in 81/113 (72%) patients, with median MFI of 28 (range: 17–153). Clinical and biological features of patients according to BCL-2 expression are summarized in Table 1: BCL-2+ patients have lower white blood cell (WBC) count at diagnosis and higher frequency of unfavorable cytogenetic-molecular risk. No differences between BCL-2+ and BCL-2- patients were observed for age, FLT3 and NPM1 mutations, for CD34 or CD56 expression, or incidence of secondary AML.

### 3.2. Complete Remission

CR was obtained in 39/113 patients (34.5%), while 68 patients (60%) were primary refractory, and 6 patients (5.5%) died during induction. Probability to achieve remission was not significantly associated with BCL-2 expression, as CR in BCL2+ patients was attained in 28/81 patients (34.5%) compared to 11/32 BCL-2- patients (34%; *p* = 0.90), nor by therapy (CR for 3 + 7 15/51, 29.5% compared to 24/62, 38.5% for HMAs; *p* = 0.40). Considering the impact of BCL2 expression according to therapy, we found a trend towards a superior CR rate in BCL2+ patients treated with HMAs (20/45, 44.5%) compared to 3 + 7 (8/35, 23%; *p* = 0.09); in BCL2- patients CR rate was similar for HMAs (4/16, 25%) and 3 + 7 (7/16, 43.5%; *p* = 0.46) (Table 2).

### 3.3. Overall Survival

In the entire population, OS at 1 year was 39%. In univariate analysis, factors associated with a shorter survival were higher in WBC count at diagnosis (12% if ≥ 30 × 10^9^/L vs. 45% if < 30 × 10^9^/L, *p* = 0.05) and ELN cytogenetic-molecular risk (48% in low vs. 43% in intermediate vs. 32% in high, *p* = 0.01); both factors retained their statistical significance also in multivariable analysis, with RR = 2.2 (1.1–4.4) for ELN risk (*p* = 0.02) and RR = 2.0 (1.1–3.3) for WBC count (*p* = 0.03). BCL-2 expression per se had no impact on survival, ad 1-year OS was 42% in BCL-2 positive patients and 31% in BCL-2 negative patients. However, considering survival with different therapies by BCL-2 expression, in BCL-2+ patients 1-year OS was 56% with HMAs and 25% with 3 + 7 (*p* = 0.02) (Figure 1a) the advantage for HMA over chemotherapy in BCL2+ patients was confirmed at 2 years (22% vs. 12%, *p* = 0.02). In BCL-2- patients 1-year OS was 36% for HMAs and 27% for 3 + 7 (*p* = 0.3) (Figure 1b). The other factor impacting on OS in BCL2+ patients was WBC count (8% vs. 48%, *p* = 0.02); however, by multivariable analysis, only therapy with HMAs was associated with longer OS in BCL2+ patients, with RR = 0.56 (0.33–0.95) (*p* = 0.03).

## 4. Discussion

Despite the significant improvement in life expectancy of young people with AML in the past decades, the outcome in elderly people remains disappointing, as long-term survival is only around 10% [8]. The increased survival in younger patients has probably been due more to the advances in supportive care and the extensive use of allogeneic hematopoietic cell transplantation than from new drugs. However, the therapeutic landscape in AML has been recently enriched with the approval of various small molecules with molecular targets, virtually suitable also for elderly and unfit patients. The evidence that BCL-2 is expressed in almost all AML cells and that, in young patients, its overexpression predicts lower CR rate and worse survival [2,9,10,11] offers an attractive option in patients ineligible for intensive chemotherapy [12,13,14]. The selective inhibition of BCL-2 by venetoclax, in combination with low-dose cytarabine or HMAs, improved the response rate and OS compared to HMAs-treated historical controls [15,16]. Nevertheless, the subsets of elderly patients taking advantage from the venetoclax-HMAs combination has yet to be established.

In the present work we first evaluated the impact of BCL-2 overexpression on disease characteristics and response to different induction regimens in a consecutive series of elderly AML patients. Increasing age, lower WBC count and unfavorable cytogenetic-molecular abnormalities were associated with BCL-2 overexpression. According to response to treatment, our data does not confirm the negative impact of BCL-2 overexpression on CR and OS, that has been recently reported by De Haes et al. in 93 AML patients treated with standard intensive chemotherapy or HMAs [4]. In line with our finding, Zhou and colleagues failed to correlate BCL-2 higher expression with worse outcomes in AML [17], but their study included both young and elderly patients. Kulsoom et al. analyzed expression of Bax and BCL-2 in 90 AML patients treated with conventional chemotherapy, without finding any significant correlation among protein expression or ratio and CR, relapse risk and OS [18].

Evaluating response to intensive chemotherapy or HMA according to BCL-2 expression, in BCL-2 positive cases HMAs were associated with a trend toward a higher CR rates and significantly longer survival compared to 3 + 7. Possible explanations of this finding may be the favorable toxicity profile of HMAs as compared to conventional chemotherapy but also to a positive effect in the “elderly” setting, in which there is increasing evidence that AML develops in a cellular context of prominent epigenetic alterations and changes in chromatin structure [19]. If this hypothesis is confirmed, the combination of HMA with venetoclax could be particularly appealing in BCL-2 overexpressing patients, due to simultaneous BCL-2 inhibition, reversion of its expression, and possibly to repression of concurrent expression of compensatory MCL-1 and Bcl-xL proteins. Recently Lachowiez et al. reported very favorable responses to the venetoclax + HMAs combination in elderly patients with AML carrying NPM1 mutations, suggesting that this regimen could be an “optimal mutation-targeted treatment approach” in this subset, irrespective of BCL-2 intensity of expression [20]. However, in vitro evidence of a correlation between venetoclax sensitivity and level of BCL-2 expression [4] suggest the potential role of BCL-2 expression as predictive biomarker. Considering the post-translational regulation of all BCL-2 family members, western blot analysis cannot be considered the best approach, which could be the semi-quantitative flow cytometric evaluation of BCL-2, possibly associated with Bcl-xL measurement.

The therapeutic landscape for AML in the elderly seems to have been profoundly changed by the advent of venetoclax, which has been associated with HMAs, low dose cytarabine, or even conventional 5 + 2 chemotherapy, with promising results [6,13,21]. Nonetheless, the most appropriate partner to venetoclax and the subsets of elderly patients taking greatest advantage from the anti-BCL-2 combination has yet to be established. Our data suggest that in BCL-2 positive cases, HMAs were associated with superior survival compared to 3 + 7 regimen. If this is due to the reduced toxicity of HMAs compared to chemotherapy, a more “targeted” effect on the epigenetic alteration characteristic of AML in the elderly has still to be defined. Furthermore, mechanistic studies are needed to investigate the reasons of possibly enhanced effect of HMAs in BCL-2 overexpressing cells. However, the combination of HMA with venetoclax seems particularly appealing in BCL-2 overexpressing patients. The rapid and simple semi-quantitative flow cytometric evaluation of BCL-2 may prove useful to predict the patients with the higher probability of response.

## Figures and Tables

**Figure 1 jcm-10-05096-f001:**
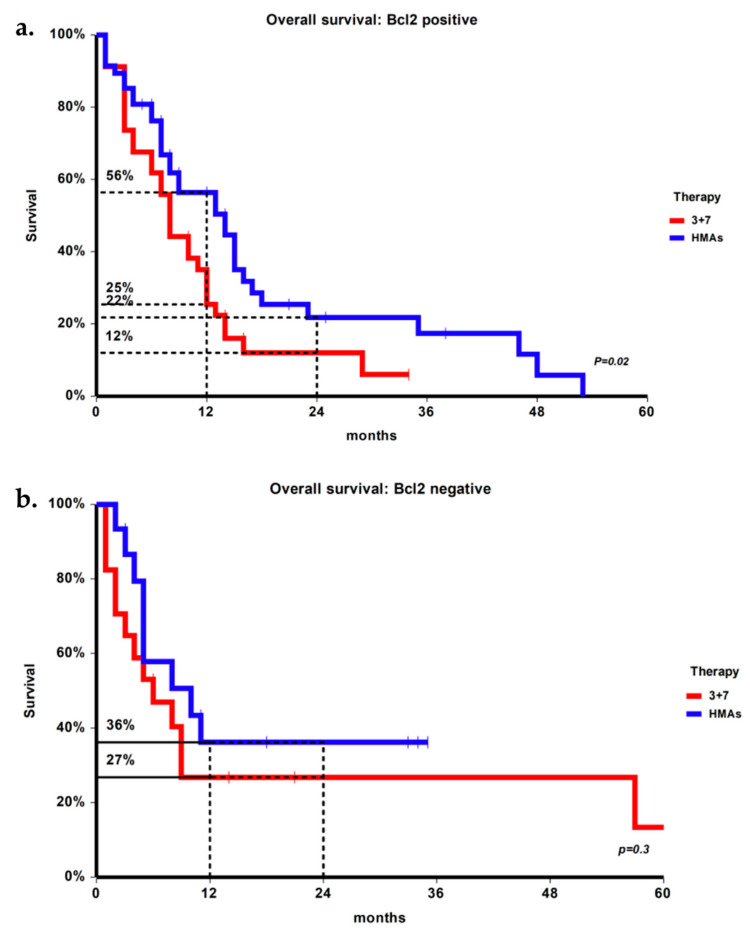
(**a**) Overall survival of BCL-2 positive patients according to therapy; (**b**) overall survival of BCL-2 negative patients according to therapy.

**Table 1 jcm-10-05096-t001:** Patients’ characteristics according to BCL-2 expression.

	BCL-2+ (*n* = 81)	BCL-2- (*n* = 32)	*p*
Median age (range), years	73 (65–85)	74 (65–83)	0.85
AML type			0.15
De novo	31	17	
Secondary	50	15	
Mean WBC (IQR), ×10^9^/L	6.6 (4.3–9.5)	8.5 (4.0–33.0)	0.048
ELN cytogenetic/molecular risk			0.045
Favorable	11	6	
Intermediate	29	18	
Unfavorable	41	8	
FLT3			0.96
ITD/TKD	13	5	
WT	68	27	
NPM			0.31
Mutated	13	4	
WT	67	26	
NA	1	2	
CD34			0.90
Positive	65	26	
Negative	16	6	
CD56			0.09
Positive	22	14	
Negative	59	18	

AML: acute myeloid leukemia; WBC: white blood cell; IQR: interquartile range; ELN: European Leukemianet; WT: wild type; NPM: Nucleophosmin; NA: not available.

**Table 2 jcm-10-05096-t002:** CR rates according to BCL-2 expression and induction therapy.

	CR	Failure	*p*
BCL-2+	28	53	0.98
BCL-2-	11	21	
3 + 7	15	36	0.30
HMA	24	38	
BCL-2+ with 3 + 7	8	27	0.05
BCL-2+ with HMA	20	26	
BCL-2- with 3 + 7	7	9	0.26
BCL-2- with HMA	4	12	

## Data Availability

The data that support the findings of this study are available from the corresponding author, up-on reasonable request.

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
