# Peer review of "BCL-2 Expression in AML Patients over 65 Years: Impact on Outcomes across Different Therapeutic Strategies"

_jcm, 2021, doi:10.3390/jcm10215096_

Round 1

Reviewer 1 Report

The manuscript from Tiribelli et al. is a retrospective analysis of 65+ yo AML patients, focused on the effect of BCL-2 expression at the time of diagnosis on OS prognosis in context of different treatment protocols.

The biggest weakness of this manuscript is, in my opinion, lack of supporting data.

In a manuscript that operates with flow cytometry analysis, I’d expect at least a representative data of a measurement, incl. gating examples and sample specification (what exactly is defined as “leukemic cells”, are they pre-isolated or not…), so the reader can be sure of what he’s looking on.

Without this, I don’t think the manuscript is ready for publication.

A few additional minor comments & questions:

  1. in introduction, there seems to be a missing word: "...and is associated with chemoresistance and shortened overall (2,3)”
  2. did they follow up Bcl-2 expression later in time, or is there really only one datapoint for each patient?
  3. did the authors differentiate the OS of primary and secondary leukemia patients?
  4. the graphs of OS, while showing some promise, are not really convincing. I feel like the authors selected a datapoint where the difference was the highest and reported that; I get that, but I’d like to see it discussed in the manuscript.

Author Response

The manuscript from Tiribelli et al. is a retrospective analysis of 65+ yo AML patients, focused on the effect of BCL-2 expression at the time of diagnosis on OS prognosis in context of different treatment protocols.

The biggest weakness of this manuscript is, in my opinion, lack of supporting data.

In a manuscript that operates with flow cytometry analysis, I’d expect at least a representative data of a measurement, incl. gating examples and sample specification (what exactly is defined as “leukemic cells”, are they pre-isolated or not…), so the reader can be sure of what he’s looking on.

Without this, I don’t think the manuscript is ready for publication.

We thank the reviewer for this comment. Method section has been developed, also in accordance with suggestions from reviewers 2 and 3, and we added an exemplary figure (marked as suppl. Fig 1).

A few additional minor comments & questions:

In introduction, there seems to be a missing word: "...and is associated with chemoresistance and shortened overall (2,3)”

The word “survival” has been added.

Did they follow up Bcl-2 expression later in time, or is there really only one datapoint for each patient?

Besides being evaluated at diagnosis in all patients, BCL-2 was tested in a minority of cases at time of relapse.

Did the authors differentiate the OS of primary and secondary leukemia patients?

In the preliminary analysis we tested if primary or secondary AML had different OS, without finding any significant difference across the various treatment strategies.

The graphs of OS, while showing some promise, are not really convincing. I feel like the authors selected a datapoint where the difference was the highest and reported that; I get that, but I’d like to see it discussed in the manuscript.

Considering the relatively poor outcome of our patients, with a 1-year survival of 39% in the entire population, we had chosen the 1-year timepoint as representative. However, superior OS with HMAs in BCL-2+ patients was confirmed at 2 years (22% vs 12%, p=0.02); this data has been added in the result section. 

Reviewer 2 Report

The role of BCL-2 in AML is still a subject of intensive studies. The correlation between BCL-2 overexpression and AML prognosis and patient survival is still being determined. The heterogeneity of AML is not without significance. Therefore studies regarding the role of BCL-2 in therapy of AML is constantly needed. The authors of submitted article aimed to investigate the impact of BCL-2 expression on the outcome of a cohort of elderly AML patients, treated with intensive chemotherapy or HMAs. Overall the paper is well-written and comprehensive, easy to follow. The results and conclusion provided by the authors add new important information on the importance of BCL-2 expression in AML. Just a few comments:

  • please provide information about the methods used to determine the molecular characteristics (ie FLT3, NPM) and antigen expression.
  • the discussion can be enriched by additional references eg: Karakas at al. Annals of Oncology 1998, 9: 159-165; Kulsoom et al. Cancer Management and Research, 2018,10: 403—416

Author Response

The role of BCL-2 in AML is still a subject of intensive studies. The correlation between BCL-2 overexpression and AML prognosis and patient survival is still being determined. The heterogeneity of AML is not without significance. Therefore studies regarding the role of BCL-2 in therapy of AML is constantly needed. The authors of submitted article aimed to investigate the impact of BCL-2 expression on the outcome of a cohort of elderly AML patients, treated with intensive chemotherapy or HMAs. Overall the paper is well-written and comprehensive, easy to follow. The results and conclusion provided by the authors add new important information on the importance of BCL-2 expression in AML. Just a few comments:

Please provide information about the methods used to determine the molecular characteristics (ie FLT3, NPM) and antigen expression.

Method section has been developed, also in accordance with suggestions from reviewers 1 and 3.

The discussion can be enriched by additional references eg: Karakas at al. Annals of Oncology 1998, 9: 159-165; Kulsoom et al. Cancer Management and Research, 2018,10: 403—416

We included the 2 suggested papers in the discussion section.

Reviewer 3 Report

The paper “BCL-2 Expression in AML Patients over 65 Years: Impact on 2

Outcomes across Different Therapeutic Strategies” intend to study the impact of BCL-2 expression on survival in the aged group. The study lacks novelty and does not provide adequate data to support the role of testing BCL2 expression in clinical decision making. The following are my review suggestions.

Major:

  1. The methodologies warrant further clarification. The patients are classified according to ELN groups, but you did not provide all of the mutational data used by the ELN group, such as ASXL1, TP53, FLT3-ITD allelic ratio. Are the secondary AML patients treated with HMA during the MDS stage? Are there any patients received allogeneic transplant? In your multivariate analysis, please indicate which parameters you put in to generate the data.
  2. The flow data of BCL2 expression should be validate by other methods, such as IHC stain for BCL2.
  3. You mention that therapy with HMAs was the only factor associated with longer OS in BCL-2+ AML. The result could be explained by the fact that BCL2+ patients have more unfavorable cytogenetics. As
  4. As for response evaluation, apart from CR and failure, are there any patients who attained CRi, MLFS, or HI after the treatment?
  5. Additional experiments to demonstrate why HMA works better in BCL2 positive patients are need. Additional data from cell line data, primary blast samples should be added.

Minor:

  1. Since your results did not include patients treated with BCL2 inhibitor. Discussion about BCL2 inhibitor should be deleted or greatly reduced.

Author Response

The paper “BCL-2 Expression in AML Patients over 65 Years: Impact on Outcomes across Different Therapeutic Strategies” intend to study the impact of BCL-2 expression on survival in the aged group. The study lacks novelty and does not provide adequate data to support the role of testing BCL2 expression in clinical decision making. The following are my review suggestions.

Major:

The methodologies warrant further clarification.

Method section has been developed, also in accordance with suggestions from reviewers 1 and 2.

The patients are classified according to ELN groups, but you did not provide all of the mutational data used by the ELN group, such as ASXL1, TP53, FLT3-ITD allelic ratio.

We agree with the reviewer’s comment. In fact, as “additional” mutational data (e.g. ASXL1, RUNX1, TP53) were not available for all the patients, some of whom have been diagnosed in 2010, we used conventional cytogenetics, FLT3 and NPM1 mutational status to define AML risk; de facto, this is in line with 2010 ELN recommendations and the validation by Mrozek et al. (JCO 2012). We modified the text and the reference list accordingly.

Are the secondary AML patients treated with HMA during the MDS stage?

None of the patients included received HMA therapy at the MDS stage. This has been specified in the text.

Are there any patients received allogeneic transplant?

None of the patients included underwent alloHCT. This has been specified in the text.

In your multivariate analysis, please indicate which parameters you put in to generate the data.

All variables with P<0.1 in univariate analysis (CD34, WBC, chemotherapy protocol, secondary AML) were included in the analysis. A backward procedure was employed to generate the final model.

The flow data of BCL2 expression should be validate by other methods, such as IHC stain for BCL2.

In a pre-analytic phase of our work, IHC has been employed to confirm flow cytometric expression of BCL2 in cell lines and in a small cohort of leukemic samples. Flow cytometry was preferred for the possibility to generate semi-quantitative data (by MFI calculation) and not only a qualitative evaluation.

You mention that therapy with HMAs was the only factor associated with longer OS in BCL-2+ AML. The result could be explained by the fact that BCL2+ patients have more unfavorable cytogenetics.

Both in univariate and multivariable analyses, ELN risk did not affect survival in BCL2+ patients.

As for response evaluation, apart from CR and failure, are there any patients who attained CRi, MLFS, or HI after the treatment?

We focused on CR as it is the strongest predictor of outcome and it is usually used, in our clinical practice, to decide if continue therapy.

Additional experiments to demonstrate why HMA works better in BCL2 positive patients are need. Additional data from cell line data, primary blast samples should be added.

We agree with the reviewer that “mechanistic” studies are needed to investigate the potential effect of HMAs in BCL2 overexpressing AML cells. We are planning experiments as a future development on our research on AML. A sentence on this point has been added in the final part of the Discussion.

Minor:

Since your results did not include patients treated with BCL2 inhibitor. Discussion about BCL2 inhibitor should be deleted or greatly reduced.

Discussion section has been revised; despite our cohort did not include venetoclax-treated patients, we feel that our finding of a superior efficacy of HMAs in BCL2+ patients could reinforce the clinical evidence of the optimal synergy of HMAs and BCL-2 inhibitors.